# The Liver and the Hepatic Immune Response in *Trypanosoma cruzi* Infection, a Historical and Updated View

**DOI:** 10.3390/pathogens10091074

**Published:** 2021-08-25

**Authors:** Natalia Vacani-Martins, Marcelo Meuser-Batista, Carina de Lima Pereira dos Santos, Alejandro Marcel Hasslocher-Moreno, Andrea Henriques-Pons

**Affiliations:** 1Laboratório de Inovações em Terapias, Ensino e Bioprodutos, Instituto Oswaldo Cruz, Fundação Oswaldo Cruz, Rio de Janeiro 21041-361, Brazil; nati.vacani89@gmail.com (N.V.-M.); carinalp@ioc.fiocruz.br (C.d.L.P.d.S.); 2Depto de Anatomia Patológica e Citopatologia, Instituto Fernandes Figueira, Fundação Oswaldo Cruz, Rio de Janeiro 22250-020, Brazil; marcelomeuser@gmail.com; 3Instituto Nacional de Infectologia Evandro Chagas, Fundação Oswaldo Cruz, Rio de Janeiro 21041-361, Brazil; alejandro.hasslocher@gmail.com

**Keywords:** *Trypanosoma cruzi* infection, Chagas disease, liver, hepatic immune response

## Abstract

Chagas disease was described more than a century ago and, despite great efforts to understand the underlying mechanisms that lead to cardiac and digestive manifestations in chronic patients, much remains to be clarified. The disease is found beyond Latin America, including Japan, the USA, France, Spain, and Australia, and is caused by the protozoan *Trypanosoma cruzi*. Dr. Carlos Chagas described Chagas disease in 1909 in Brazil, and hepatomegaly was among the clinical signs observed. Currently, hepatomegaly is cited in most papers published which either study acutely infected patients or experimental models, and we know that the parasite can infect multiple cell types in the liver, especially Kupffer cells and dendritic cells. Moreover, liver damage is more pronounced in cases of oral infection, which is mainly found in the Amazon region. However, the importance of liver involvement, including the hepatic immune response, in disease progression does not receive much attention. In this review, we present the very first paper published approaching the liver’s participation in the infection, as well as subsequent papers published in the last century, up to and including our recently published results. We propose that, after infection, activated peripheral T lymphocytes reach the liver and induce a shift to a pro-inflammatory ambient environment. Thus, there is an immunological integration and cooperation between peripheral and hepatic immunity, contributing to disease control.

## 1. Introduction

### 1.1. The History of Chagas Disease and the Description of the Trypanosoma cruzi Protozoa

In 1907, Dr. Carlos Chagas (1879–1934), a researcher who had recently graduated in medicine, was sent by Dr. Oswaldo Cruz, his mentor, to the interior of Brazil. The Central do Brasil railway was being built in the north of the state of Minas Gerais, and he was designated to fight malaria, a plague that was affecting most workers. A railroad engineer brought to Dr. Chagas’ knowledge that a blood-sucking insect inhabited the very poor huts in the area in great numbers [1]. At that time, he was not aware that those insects could be a vector for any disease. Still, he wrote later that “a uniform morbid (clinical) condition immediately came to our attention, appreciable in almost all children in the area where the invertebrates abounded… however, sometimes observed in adults” [2]. When he started studying these insects that usually fed on the peoples’ faces at night, Dr. Chagas wrote in his full paper published in 1909 that they “were found in immense numbers in the cracks of the walls, which were not plastered” [3]. Dr. Belizario Penna was with Dr. Chagas when the first insects were collected for microscopic analysis in a train car, which functioned as an improvised laboratory. Dr. Penna wrote, “... spending the night at a home…, where I was able to collect a large number of insects suctioning in children... it was possible to see his uproar when he discovered in the blood taken from the insects’ intestines a flagellated parasite” [4].

Dr. Chagas first observed the parasite when he analyzed samples from the insects’ intestinal posterior portion in loco, where there were “numerous flagellates.” Then, insect specimens were sent to Dr. Oswaldo Cruz for analysis at the Instituto Oswaldo Cruz, previously Federal Serotherapy Institute, located in Rio de Janeiro, Brazil. It was observed that 20 to 30 days after the insects fed on marmoset monkeys (*Callithrix penicillata*), numerous parasites were found in their blood. The protozoan was identified as having a morphology entirely different from that of any known species of the genus Trypanosoma, and the new parasite species was named *Trypanosoma cruzi* in honor of Dr. Oswaldo Cruz. Dr. Chagas also reported: “... *Trypanosoma cruzi* could be identified, it is undoubtedly a new and extraordinarily strange human trypanosomiasis, which is caused by a parasite which offers very interesting phases in its developmental circle.” [2,5]. However, after reanalyzing blood forms of the parasite in a posterior published paper, Dr. Chagas wrote that “…the very particular development of this protozoan ... seemed to justify the creation of a new genus, which we did, changing the name to *Schizotrypanum cruzi*.” [3]. As we know today, his first impression was correct, and the genus *Schizotrypanum* is no longer used. Three different forms are found in the parasite’s life cycle, and all of them were described by Dr. Chagas. He observed epimastigote forms in the insects’ intestines, trypomastigote forms in the blood of patients and others hosts, and (intracellular) amastigote forms in vertebrate lungs.

Once Dr. Chagas identified infected insects that could transmit the infection to mammals in the laboratory, he studied whether those parasites would be found in the blood of individuals and domestic animals residing in insect-infested houses. Shortly before identifying the first human case in a two-year-old child named Berenice, Dr. Chagas visualized *T. cruzi* in a domestic cat [3]. Then, he reported that he obtained venous blood from a child and that the blood was injected into guinea pigs that survived for only six days. During the necropsy, abundant parasites were found in the lungs of the animals.

Dr. Carlos Chagas, in his 1909 study of the disease’s acute phase , described the most prominent clinical signs and symptoms as “…great anemia, marked organic decay, sub-eyelid edema and often generalized edema, ... bulky ganglia ..., fever” among others. In addition, he described frequent and prominent splenomegaly in acutely infected residents, which could not be attributed to malaria, and hepatomegaly. A few years later, in 1912, Dr. Chagas reported that the new protozoan was observed in an armadillo, the first sylvatic reservoir host [1]. Many different reservoir species would be gradually described, providing evidence for an enzootic cycle of *T. cruzi*. 

Many other scientists played an important role in describing various aspects of the disease and its pathogenesis. In his first published paper in 1909, Dr. Chagas himself acknowledged Dr. Oswaldo Cruz for his invaluable contribution and mentoring. He also mentioned Dr. M. Hartmann and Dr. Stanislaus von Prowazek. The latter was a Czech zoologist and parasitologist that spent about six months working at the Federal Serotherapy Institute with Dr. Oswaldo Cruz. Moreover, Dr. Gaspar de Oliveira Vianna (1885–1914) dedicated himself to studying the pathological anatomy of Chagas disease and verified new facts in the evolutionary cycle of *Trypanosoma cruzi* [6]. Dr. Arthur Neiva (1880–1943) was an entomologist and studied the genera Triatoma, which includes some of the invertebrate vectors of the Chagas disease, making important contributions to the parasite’s biological cycle description in the invertebrate host [7].

Regarding the diagnosis of Chagas disease, Guerreiro and Machado developed a serological test based on complement fixation [8], just as Emile Brumpt introduced xenodiagnosis [9]. Magarino Torres described how the parasite infects the individuals, identifying the vector’s defecation after a repast as the primary mechanism of infection [10].

When describing the first human case of Chagas disease, at that time still named American trypanosomiasis, Dr. Chagas became the first physician and scientist to describe an infectious disease in its entirety. He introduced all the elements involved to the scientific and medical community: the causal agent (*Trypanosoma cruzi*); the vector (*Triatoma infestans*); wild and domiciled reservoirs (vertebrate animals); the clinical signs and symptoms; the diagnosis; the prognosis; the epidemiology; and the ecology with all the dynamics of the disease transmission.

Dr. Chagas identified and described in his initial work, and those published in the next few years, only the acute phase. The symptomatic chronic phase, mainly characterized by cardiac disease, was later described by Dr. Chagas in partnership with Dr. Eurico Villela in 1922 [11]. In another study, Dr. Eurico introduced the first graphic records using polygraphs of the heart disease, and updated the diagnosis methods [12]. In the 1930s, Dr. Evandro Chagas, the older son of Dr. Carlos Chagas, reviewed the Chagas disease in its etiological, anatomical, clinical, and therapeutic aspects. Moreover, Dr. Evandro showed the first electrocardiograms and chest X-ray in Chagas’s heart disease [13,14]. In the very first note published by Dr. Chagas [2], he mentions that “…repeated blood exams (for circulating parasites) in children with the chronic condition were negative”. Identifying the recent chronic phase is particularly remarkable; once clinical symptoms are entirely different from the acute phase, only about 30% of infected individuals will develop cardiac symptoms, usually decades after the acute phase, and blood parasitemia is subpatent.

### 1.2. Epidemiology and Clinical Aspects of Chagas Disease

Chagas disease was described in the interior of Brazil, but the triatomine vector species that transmit the disease are geographically distributed throughout Latin America and today include the southern United States. Because of migratory movements, the disease is found on almost every continent, in countries such as the USA, Spain [15], Japan, Australia, France, and others. According to the World Health Organization, six to seven million people are infected worldwide, although most cases are found in Latin America [16]. The main transmission route is through triatomine bugs, which defecate while sucking blood from the host. The feces contain the metacyclic trypomastigote forms, which invade host cells, differentiate into amastigote forms, and multiply by binary division. This intracellular form differentiates back into trypomastigote forms in the cytoplasm, disrupts the host cell, and infects neighboring cells or is carried by the circulation (Figure 1). Over the past few decades, the risk of infection has been reduced due to vector control campaigns and other strategies to mitigate vector-borne and blood transmission in many endemic countries. However, several challenges have hampered the effective implementation of disease surveillance due to new outbreaks of orally transmitted Chagas disease in some countries and the possibility of vertical transmission even in nonendemic areas.

Usually, acutely infected adult patients are asymptomatic or present with mild and nonspecific symptoms, such as fever, headache, and general malaise. In some cases, acute myocarditis is observed, which may be associated or not with meningoencephalitis, which is particularly fatal in children. After about three months, the chronic phase initiates, and blood parasites are no longer observed upon microscopic examination, although with persistent positive serology for life. This phase starts with the indeterminate form, a disease stage where clinical evaluation, chest radiography, electrocardiogram, and contrast-enhanced esophageal and colon examinations are normal. The indeterminate form can last for decades, and, for unknown reasons, about 30% of chronic patients will develop cardiac alterations, which frequently involve rhythm and/or conduction heart disorders, left ventricular systolic dysfunction with or without heart failure, and thromboembolic phenomena. Up to 10% will develop digestive alterations, which involve peristalsis dysfunction of the esophagus and/or intestine and megasyndrome presentations, or even cardiac and digestive mixed alterations [17,18].

In some South American countries, notably Brazil and Venezuela, the main transmission route has changed, and currently, oral infection is responsible for most acute cases, especially in the Amazon region. In those states, raw fruit juices and mashed fruit pulps are part of the diet in the local culture, and triatomines can be triturated together with the food. The acute phase has higher morbidity and mortality in oral infection, with substantial hepatic impairment and acute chagasic myocarditis of greater intensity [19]. Thus, it is essential to study the hepatic component of the disease as the epidemiological characteristics are dynamic.

### 1.3. The Founding Fathers of the Liver Pathology in Chagas Disease

Dr. Vianna in 1911 was the first to perform an anatomopathological study of the disease, and he noted a direct correlation between the level of hepatic fatty degeneration with the morbidity of acutely infected patients [6]. Although in 1912 Dr. Rocha Lima briefly mentioned that he did not find any fatty transformation or necrosis in the liver of patients, in 1916 Dr. Carlos Chagas reinforced the association of hepatic steatosis with the acute infection. Interestingly, in this work, Dr. Chagas did not find parasites infecting hepatic cells and proposed that “toxins” produced by *T. cruzi* would be responsible for the liver pathogeny [20].

In 1920, parasite-induced liver disease was studied in more detail by C. Pinheiro-Chagas, and he described that “…in the acute form of the disease, there is predominantly a fatty infiltration, more abundant in the periphery of the lobe, along with hypertrophy and hemosiderotic pigmentation in the Kupffer cells, with a small lymphomonocytic infiltrate inside the dilated intralobular or periportal capillaries” [21]. The description continued, and he wrote, “…in chronic cases, there is intense necrosis of centrilobular coagulation, characteristic of chronic (cardiac) passive congestions”. In addition, “there is conjunctiva hyperplasia and mononuclear infiltration in the portal spaces”. Analyses of chronic patients were carried out in individuals with heart disease and positive parasitological diagnoses in heart sections. These analysis are particularly extraordinary since, in the chronic phase, parasites in the cardiac tissue are extremely rare, which led in more recent times to the proposal of autoimmune myocarditis induced by the infection. However, this proposal is no longer considered because the parasite persists in the myocardium for decades.

In 1923, Dr. Crowell undertook an anatomopathological study on the liver of an eight-month-old child who died with acute Chagas disease [22]. There were already hepatomegaly and hemorrhagic foci, with fatty degeneration in the entire organ, mainly in the perilobular region and portal spaces. In 1925, Dr. Almenara studied the hepatic histological lesions of four patients who died during the acute phase, and Dr. Carlos Chagas himself supplied the samples [23]. Once again, intense fatty degeneration was evident, and he mentioned that “…hepatocytes lose trabecular disposition, and even the better-preserved cells have protoplasm loaded with fat droplets”. He also observed, “…hepatocytes with pycnotic nuclei, karyolysis, karyorrhexis, areas of necrosis, hypertrophied and hyperplasia Kupffer cells containing hemosiderin”. However, he found no parasites in the liver.

In 1938, Drs. Jôrg Mazza and Canal Feijó studied the organs of a young, chronic, 17-year-old patient who died of acute heart failure and had contracted the disease eleven years before. Her liver had “characteristics of chronic passive congestion and a cirrhotic appearance”. Between the lobes, there were circular nodules, with lymphocytic infiltration and a dense network of reticular histiocytes [24].

Dr. Margarino Torres, who greatly contributed to the description of the cardiac alterations, published with Dr. Duarte in 1948 the case of another child who died in the acute phase [25]. They observed moderate congestion of the centrilobular sinusoids and discrete perilobular fatty infiltration without intracellular parasites and rare necrotic hepatocytes. In 1949, Dr. Jairo Ramos and collaborators carried out the first specialized study known to analyze the functional changes of the liver in the disease. They separated thirty chronic patients into three groups; all had Chagasic myocarditis [26]. The individuals in group A did not have heart failure, group B had mild, and group C had severe cardiac insufficiency. The authors concluded that “the general balance of protein dosages in the blood and their correlation with liver function does not allow to state that there is defined and systematic liver failure”.

In 1954, Dr. Sadek and Dr. Edmundo Vasconcelos published an elegant work summarizing several studies addressing the hepatic alterations in Chagas disease and concluded that structural changes are different in acute and chronic patients [27]. In general terms, they described that in the acute phase, the lesions are “constituted by fatty infiltration, especially perilobular, rarely diffuse and of varying intensity; areas of necrosis are not very extensive and have no preferential location; hyperplasia and hypertrophy of Kupffer cells; vascular congestion and lymphomonocytic infiltration in intralobular or periportal capillaries “. In the chronic phase, the liver lesions observed are “those typically seen in chronic (cardiac) passive congestion, the so-called *nutmeg* liver of cardiac patients”. Since the existence of toxins produced by the parasite was not confirmed, as proposed by Dr. Carlos Chagas in 1916, Drs. Sadek and Vasconcelos proposed that “the histological lesions found in the liver of… chronic individuals are the result of heart failure,” and they added that functional liver disorders may be the result of cardiomyopathy or even food deficiency. Then, the pathophysiological processes of the acute phase are more directly related to the parasite infection. On the other hand, in the chronic phase, the hepatic alterations are due to stasis caused by decompensated heart failure, which leads to liver congestion and abdominal pain mainly caused by liver capsule distension. Therefore, the chronic alterations are not directly linked to the presence of the *T. cruzi* itself or immunological phenomena inherent to the Chagas disease.

The liver lesions found in experimentally infected animals, mainly dogs and some rodent species, are different from the lesions found in patients and “…are characterized by the presence of specific granuloma [28], necrosis and fatty infiltration, mainly centroglobular in intensity, variable hyperemia, and intracellular infection mainly in Kupffer cells” [27].

### 1.4. The Participation of the Liver in T. cruzi Infection, a More Recent Perspective

Today we know that the liver plays an essential role in the infection, and its importance in parasite clearance and destruction of blood (trypomastigote) forms, for example, is well documented. Experimental murine infections showed that specific antibodies against the parasite plus phagocytic cells are required for extracellular trypomastigote clearance. IgG-coated parasites are phagocytosed by resident mononuclear cells, especially in the liver but also in the lungs and spleen [29]. Moreover, pro-inflammatory cytokines, mainly IFN-γ, potentiate the trypanocidal activity of the phagocytic cells [30], and intact Fc portions of IgGs are required [31]. Sardinha and cols. published in 2010 that the liver is the primary site of parasite accumulation just one hour after intravenous injection of *T. cruzi* trypomastigote forms in chronically infected mice [32]. At this time point, viable parasites and parasite remnants were observed scattered in the liver parenchyma, which considerably diminished after 48 hours, and no intracellular parasites were observed in the liver seven days after the challenge [32]. Moreover, transmission electron microscopy showed platelet thrombi occluding small vessels in the lung, liver, and spleen, and phagocytosed parasites in different stages of destruction were found within macrophages, neutrophils, and eosinophils. Therefore, it seems that not a particular cell population, but different cells, act in concert to destroy the parasites in the liver [29]. 

The liver is the main synthesis site for the complement system’s components, and it has long been evaluated if this lytic pathway could play a role in removing blood parasites. Although trypomastigote clearance is dependent on C3 [33], it is primarily independent of the lytic terminal pathway. A more detailed analysis of the complement system’s importance in parasite clearance showed that C1q, C3, mannan-binding lectin, and ficolin molecules bind to trypomastigote forms. Moreover, C3b and C4b deposition assays revealed that *T. cruzi* activates mainly the lectin and alternative complement pathways in non-immune human serum [34]. Experiments using C5-deficient mice showed no difference in parasite clearance compared with wild-type mice [31].

It is long known that blood trypomastigote forms express several complement system inhibitors, such as a decay-accelerating factor expressed by *T. cruzi* (T-DAF) [35], complement C2 receptor inhibitor trispanning [36], complement regulatory protein, and others [37]. Nevertheless, some parasite strains seem to be susceptible to the complement system [34].

It is also known that the infection subverts the host lipid metabolism in multiple ways [38], mainly affecting the low-density lipoprotein- (LDL) and high-density lipoprotein-dependent (HDL) pathways and their receptors. LDL is generated from liver-derived very-low-density lipoprotein (VLDL) and is a potent inhibitor of *T. cruzi* trans-sialidase, an enzyme expressed mainly by epimastigote and trypomastigote forms of *T. cruzi* that transfers sialic acid from the environment to the parasite surface [39]. Moreover, the addition of LDL and HDL to cells in culture enhances infection by *T. cruzi* trypomastigote forms in a dose-dependent manner [40]. The LDL receptor (LDLr) is one of the molecules used by the parasite during cell invasion, and in vitro infection in the presence of an LDLr blocker resulted in a 42% reduction of intracellular infection [41]. The LDLr is also involved in the trafficking of lysosomes to the cytoplasmic parasitophorous vacuole, and the disruption of the LDLr pathway affects the intracellular parasite load. 

Today we know some of the lipid metabolic pathways that led to the first observations of hepatic steatosis by Dr. Chagas. For example, it was recently demonstrated that *T. cruzi* interaction with LDLr leads to the accumulation of LDL cholesterol in host tissue in acute and chronic chagasic patients [42]. Moreover, murine experimental infection revealed a significant increase in the absolute amount of triacylglycerides, cholesterol, and cholesterol esters in liver microsomal membranes [43]. Additionally, *T. cruzi* experimental infection was considered a potent risk factor for non-alcoholic steatohepatitis, associated with strong oxidative stress and metabolic disorders [44].

Regarding the hepatic immune response, we have just recently started to understand the integration and possible interdependency between hepatic and peripheral immunity after infection. As primarily observed at the beginning of the last century, the *T. cruzi* infection leads to inflammatory mononuclear cell infiltration in the liver parenchyma. Today we know some of the main cell types that compose the inflammatory foci and their inflammatory mediators. Briefly, as this topic will be discussed in more detail in the next section, the infection induces an increase in Mac1^+^, activated CD8^+^ and CD4^+^ T lymphocytes expressing CD25, CD69, and/or CD122, natural killer (NK), and NKT cells [32] in the liver. Moreover, one of the significant roles of NK cells and CD4^+^ T lymphocytes in liver protection and infection control is interferon-gamma production (IFN-γ) [45]. In addition, we observed that experimental murine infection leads to increased hepatic regulatory T (Treg) cell numbers, higher expression of programmed death ligand 1 (PD-L1 or B7-H1) in the liver stroma, increased blood activity of ALT and AST transaminases, and other alterations [46].

### 1.5. The Immune Response in the Liver

The liver is the second largest organ in the human body. It performs many essential functions, including metabolic regulation, digestion, production of bile, detoxification (conjugations with sulfate, glucuronic acid, glutathione, acetate, and glycine), and biotransformation of drugs and toxins (oxidations-reductions and hydrolysis) [47,48,49]. Approximately 80% of the blood supply that reaches the liver comes through the hepatic portal vein, consisting of blood that is low in oxygen and rich in nutrients and molecules of the intestinal microbiota. This anatomical characteristic determines that the liver typically meets very high levels of bacterial components that, in the periphery, would be recognized as danger signals and potent pro-inflammatory stimuli. The liver must then be able to individually discriminate pathogenic damage-associated molecular patterns (DAMP) [50] and especially pathogen-associated molecular patterns (PAMP) [51] from harmless DAMPs and PAMPs. This means that the liver must retain its tolerogenic bias and selectively recognize proper danger signals for pro-inflammatory response against infections or tumors, for example. These fascinating properties are just beginning to be elucidated.

Many anatomical, immunological, and environmental aspects play a central role in balancing tolerance versus immune responses in the liver. For example, the organ is highly vascularized, and these hepatic microvessels are known as hepatic sinusoids. The epithelial cells that line the sinusoids are fenestrated, allowing the protrusion of membrane segments and physical interaction between cells flowing in the vase lumen with stromal and parenchymal cells. Between the sinusoids wall and hepatocyte cords, there is the Disse space [52], a space adjacent to the sinusoids that harbors many different cell types in the liver. The very low blood pressure in the sinusoids favors this integrated cellular interaction network, affecting the liver’s biochemical and immunological functions [53]. Although the liver is best known for its primary metabolic functions, the organ is of great importance in the local and systemic immune response [54,55], as depicted below. 

#### 1.5.1. Resident Liver Cells

##### Hepatocytes

Hepatocytes compose approximately 60% of the hepatic cells and about 90% of the liver volume [53], and their primary function is formation and excretion of bile; lipid synthesis and plasma lipoprotein secretion; control of cholesterol metabolism; regulation of carbohydrate homeostasis; formation of urea, serum albumin, coagulation factors, enzymes, and other molecules; and metabolism or detoxification of drugs and other exogenous substances [56].

As mentioned before, peripheral cells recirculating in the sinusoids’ lumen can project filopodia through the fenestrae and the Disse space and directly interact with hepatocytes, which serve as antigen-presenting cells (APCs). As hepatocytes express MHC-I, they can prime naïve CD8^+^ T lymphocytes either with endogenous antigens or via cross-presentation [57]; however, hepatocytes fail to provide activated T lymphocytes with the required survival factors and lead to CD8^+^ T lymphocyte deletion [58,59]. Under steady-state conditions, hepatocytes do not express MHC-II and may not lead to CD4^+^ T lymphocyte activation (Figure 2). However, it was observed in clinical hepatitis (viral or autoimmune) that hepatocytes often exhibit aberrant MHC class II expression [60]. Although in vitro assays using transgenic MHC-II^+^ hepatocytes showed that CD4^+^ T lymphocytes were activated, this capacity remains to be conclusively demonstrated in vivo.

In acute chagasic patients, most likely infected by the oral route, levels of hepatic transaminases and activated C protein were increased, although with lower levels of coagulation factor VII [61]. All these proteins are synthesized by hepatocytes and indicate the profound impact of the infection on this cell type. However, hepatocytes are not commonly observed to be infected in vivo.

##### Kupffer Cells

Kupffer cells (KC) represent the largest population of mononuclear phagocytes in the body and account for 20 to 30% of non-parenchymal cells in the liver. KCs are derived from a self-renewing pool of organ-resident stem cells originated from the fetal yolk sac and bone marrow-derived monocytes [62]. These cells are located in the sinusoid lumen, mainly in the periportal area, where they contact recirculating T lymphocytes and meet PAMPs from the flora and other intestinal molecules. KCs can also project cellular segments through the Disse space and reach adjacent hepatocytes [63,64,65].

Different from circulating and tissue macrophages, hepatic KCs are more associated with the F4/80^+^CD68^+^ phenotype than F4/80^+^CD11b^+^. In those cells, CD68 expression may be related to LDL endocytosis, as CD68 is the main receptor for oxidized LDL. Moreover, oxidized LDL and its receptors may be involved in cholesterol absorption from the diet through the portal circulation. In the liver, three populations of KCs are often identified, which are F4/80^+^CD11b^−^CD68^+^, F4/80^+^CD11b^+^CD68^−^, and F4/80^+^CD11b^+^CD68^+^. CD11b^+^ KC subsets feature a greater capacity to produce cytokines, such as TNF and IL-12, than CD68^+^ KCs. However, CD68^+^ KCs show more potent phagocytic activity and production of reactive oxygen species (ROS) after lipopolysaccharide (LPS) stimulation [66,67]. In addition, a common feature in liver injury is the infiltration of circulating monocytes that increase macrophage-like cells in the organ [68]. In mice, after inflammatory stimulus, bone marrow-derived CCR2^+^ LY6C^+^ monocytes give rise to macrophages phenotypically and functionally different from resident KC [69]. When these cells mature, they down-regulate the Ly6C expression and act according to hepatic microenvironmental signals [70].

KCs express MHC-I and MHC-II and co-stimulatory molecules such as B7.1, B7.2 [71], and CD40, although at lower levels than hepatic dendritic cells (HDC). Under normal conditions, KCs secrete prostaglandin E2 (PGE2), transforming growth factor beta (TGF-β), and IL-10 [46], and express Fas-L and PD-L1 on the cell membrane [72,73] (Figure 2), playing a role in maintaining immune hyporesponsiveness/immunotolerance and leading to the differentiation of more hepatic Treg cells. 

In vivo assays showed that KCs can induce apoptosis of neutrophils and other polymorphonuclear cells (PMNC) through the Fas/Fas-L pathway [74]. However, this phenomenon was not observed when evaluating macrophages from other tissues, such as the lungs and spleen. This might be due to additional required molecules and cell populations found in the hepatic environment, and indeed it was observed that KC-dependent PMNC apoptosis depends on P-selectin expression on hepatic sinusoids. It was also demonstrated that phosphatidylserine (PS) and PS receptor are necessary for apoptotic PMNC phagocytosis by KCs [75]. Furthermore, the engagement of PS receptor induces the secretion of more TGF-β, IL-10, and PGE2 and the reduction of pro-inflammatory cytokines by KCs [76], contributing to the maintenance of liver tolerance.

##### Natural Killer Cells

The liver has an unusually high concentration of NK cells, consisting of approximately 50% of total lymphocytes in the organ, while NK cells represent 5–20% of circulating lymphocytes in humans [49,77,78,79]. These bone marrow-derived cells are components of the innate immune system, and hepatic NK cells comprise both liver resident (lr-NK) and transient conventional NK (cNK) cells [80]. The main cytokines produced by NK cells are IFN-γ, which is pivotal in protecting the liver after *T. cruzi* infection [45], and TNF that has been associated with apoptosis in the liver after infection [81]. Furthermore, NK cells have been reported to secrete cytokines such as IL-5, IL-10, IL-13, the growth factor granulocyte-monocyte colony-stimulating factor (GM-CSF), and the chemokines MIP-1, IL-8, and RANTES, among others [82,83]. 

The outcome effector function of NK cells in any tissue is defined by a balance of antagonistic activating versus inhibitory signaling molecules, collectively named the killer activation receptors (KARs) and killer inhibitory receptors (KIRs). Both groups of receptors simultaneously interact with potential target cells, and the levels of MHC-I expression will determine their fate [84]. The inhibitory receptors include the C-type lectin-like receptor NKG2A, which forms a heterodimer with CD94 (CD94/NKG2A), KIR2DL, KIR3DL, and others. These receptors recognize non-classical or classical MHC-I alleles expressed on the surface of autologous target cells and, in the presence of normal MHC-I levels, this interaction activates phosphatases such as SHP-1 and SHP-2 in the NK cell [85]. Therefore, when NK cells bind to normal cells, the phosphatases recruited by the inhibitory receptors are taken to the synapse, where they act to prevent NK cell activation and effector function against bystander cells. However, some virus infections and tumors downmodulate MHC-I expression below a threshold level, and, in this case, the inhibitory signals are surpassed by the activation of the KARS. These activating receptors, such as NKp30, NKp46, and NKp44, recognize the stress-related molecules MICA and MICB expressed by unhealthy cells, and this interaction leads to the secretion of perforin and granzymes, cytotoxic mediators that kill the target cell [80,86,87]. 

It has been published that hepatic NK cells play a role in liver regeneration [88] and immunological tolerance. It was also demonstrated that HDCs primed by NK cells via the NKG2A inhibitory receptor capacitated the DCs to induce the differentiation of more CD4^+^CD25^+^ Treg cells. NKG2A triggering also led to increased secretion of TGF-β by the NK cells, which was involved in the generation of this NK-induced type of DC. The Treg cells induced by NK-primed DCs exert their suppressive function through the negative costimulator programmed death-1 (PD-1) molecule [89].

In experimental *T. cruzi* infection, NK cells were observed to be increased up to six-fold in different mouse lineages by day seven after infection and were the main producers of IFN-γ at this time point [45].

##### Natural Killer T Cells

Natural killer T (NKT) cells are nonconventional T lymphocytes that share phenotypic and functional characteristics with NK cells. NKT cells are divided into two subgroups, which are type I and type II. Type I cells express a semi-invariant T cell receptor (TCR) and are denominated classical or invariant NKT (iNKT). Human iNKT cells express homologous Valpha24-Jalpha18 chains paired with Vbeta11 [90]. These cells can be activated by glycolipid antigens presented by the MHC-I-related CD1d molecule [90], typically expressed by APCs, and part of NKT cells in humans can express the coreceptors CD4 or CD8 [90]. Type II NKT cells are rare, representing less than 5% of liver NKT cells. They express more diverse TCRs, and may also recognize microbial phospholipids and sulfatides besides glycolipids in the context of CD1d [91,92,93]. NKT cells also express markers usually found in primed T lymphocytes, either effector or memory cells, such as CD25, CD44, CD69, and CD122.

Hepatic NKT cells seem to be important in the process of multiple cell type activation, as CD1d-deficient mice showed a significant decrease in NK, macrophage, neutrophil, and conventional B and T lymphocyte function in the liver after in vivo viral infection [94]. These cells were fully activated in control wild-type mice, and primed NKT cells could secrete IFN-γ. Therefore, NKT cells play direct and indirect roles in regulating liver injury, inflammation, fibrosis, and tumor response [95]. 

It was observed that infected CD1d^−/−^ mice, which lack type I and type II NKT cells, develop a mild infection with reduced liver mononuclear cell infiltration [96]. On the other hand, control Jalpha18^−/−^ mice, which lack only iNKT cells, have a more severe infection, and most individuals die. Thus, the authors suggest that iNKT cells dampen the inflammatory response, possibly regulating type II NKT cells that would be pro-inflammatory.

##### Hepatic Dendritic Cells

HDCs are one of the main APCs in the liver and are central in regulating the immunological balance between tolerance and pro-inflammatory responses. HDCs, like other tissue-specific DCs, are produced from hematopoietic stem cells in the bone marrow and are distributed as immature DCs into lymphoid and non-lymphoid tissues [97]. In addition to immature DCs, hematopoietic stem cells also give rise to two DC precursors, myeloid monocytes and plasmacytoid DC precursors, named pre-DCs [97,98]. Moreover, in vitro assays showed that human blood monocytes incubated with rat liver epithelial cells give rise to a DC subset that promotes a Th2-biased response [99]. 

HDCs are mostly limited to the perivenular region, portal space, and the Glisson capsule in the normal liver, but some HDCs can be scattered throughout the parenchyma. In the liver, some cytokines like fms-like tyrosine kinase 3 ligand (Flt3L) and GM-CSF can recruit DCs from the bone marrow [100]. Moreover, some anti-inflammatory and immunosuppressive drugs can affect DCs recruitment to the liver and HDC maturation and function; these include aspirin, corticosteroids, calcineurin inhibitors, and rapamycin [54,100,101]. 

In the liver, DCs are generally classified as steady-state or inflammatory cells, with some phenotypic differences between murine and human cells [102]. Under steady-state conditions, hepatic DC subsets include plasmacytoid DC (pDC) and conventional DC (cDC), which are subdivided into cDC1 and cDC2 [103]. Unlike steady-state DCs, inflammatory cells originate from classical (CD14^+^) or non-classical monocyte precursors (CD16^+^) in humans, or LyC6^high^ or LyC6^low^ monocyte precursors in mice, after inflammatory stimuli [104]. Different DC subsets play essential roles in regulating the immune response in the liver, and the cDC1 population can be identified in mice as MHC II^+^, CD11c^+^, CD103^+^, and Langerin^+^, while cDC2 cells are MHC II^+^, CD11c^+^, CD11b^+^, SIRPa^+^, and CX3CR1^+^ [105]. pDCs, on the other hand, are CD11c^+^ and sialic acid-binding Ig-like lectin H (SiglecH)^+^ and do not express conventional DC markers, such as XCR1, SIRPa, CD11b, CD24, and CD26. In addition to the phenotypic classification, other related factors aid the classification of DCs, such as ontogeny, key gene signature, expression of Toll-like receptors (TLR), C-type lectin, and chemokine profile [106]. Liver DC subsets detect infectious signals and fluctuate in frequency, as cDC1 are significantly reduced, and cDC2 are increased after hepatitis C virus (HCV) infection, for example [107]. HDCs are also distinct from extrahepatic DCs regarding the cytokines secreted, with lower levels of IFN-γ and more IL-10 than IL-12, favoring Th2 responses [108,109]. 

Under steady-state conditions, hepatic pDCs are more immature APCs, with lower endocytic capacity and lower expression of MHC-II [110] and co-stimulatory molecules, such as CD40, CD80, and CD86 [111,112] (Figure 2). The other hepatic DC subpopulations express higher levels of these molecules. However, it is important to highlight that the expression of the required molecular repertoire for antigen presentation in the liver does not necessarily mean that these cells are classically operational. For example, HDCs may express high levels of PD-L1, TGF-β, PGE2, and other downmodulation molecules that maintain the immunological tolerance observed in the liver [113] (Figure 2). 

##### Stellate Cells

Hepatic stellate cells (HSC) were described by Kupffer in 1876 using a gold chloride method for neuronal component detection in the liver; Kuppfer named them “sternzellen” (star cell, in German) [114]. HSCs represent only about 10% of the liver’s total resident cells and are located in the subendothelial Disse space [115]. HSCs are best known for their capacity to store vitamin A [116] and retinyl esters in cytoplasmic lipid droplets, which is their most distinctive feature [117]. The liver stores the majority of vitamin A, and up to 90% of the hepatic retinol is located in HSC’s lipid vacuoles [118]. It has also been demonstrated that HSCs modulate multiple phenotypic markers according to the lobular location in the liver [119,120] and are considered functional APCs in the organ. A pioneering work showed that human HSCs maintained in culture express many molecules involved in antigen presentation, including members of the HLA family (HLA-I and HLA-II), lipid-presenting molecules (CDlb and CDlc), and factors involved in T-cell activation (CD40 and CD80) [121] (Figure 2). These cells could perform receptor-mediated endocytosis and phagocytosis of bacteria, for example, and all features were markedly increased after incubation with the pro-inflammatory cytokines IL-1β and IFN-γ. Moreover, functional assays demonstrated that murine HSC can efficiently present antigens to CD1d-, MHC-I, and MHC-II-restricted T lymphocytes [122].

Although previous authors suggested that HSC are professional intrahepatic APCs that elicit many T cell responses, in contrast to other liver cells that lead primarily to tolerance, additional results should be considered. For example, it was also demonstrated that HSCs are central in the liver’s regulatory response, as they can inhibit T cell responses via B7-H1-mediated apoptosis [123]. Moreover, it was shown that HSCs alone do not present antigens to naive CD4^+^ T lymphocytes but, in the presence of dendritic cells and TGF-β, preferentially induce FoxP3^+^ Treg cells [124]. 

Under normal conditions, HSCs maintain a non-proliferative phenotype, but when exposed to pathogens or after liver damage, they become activated and can transdifferentiate from vitamin A-storing cells to collagen-producing myofibroblasts. Therefore, these cells are primarily considered to be the most important cell type in liver fibrosis [125]. Using a murine model for *T. cruzi* infection, it was observed that in vivo treatment with 15-deoxy-Δ prostaglandin J2 (15dPGJ2), a natural agonist of peroxisome-proliferator activated receptor (PPAR) γ, induced potent anti-inflammatory effects, leading to reduced pro-fibrotic cytokines, hepatic collagen deposition, cellular inflammatory infiltration, and hepatic damage [126]. However, the authors did not investigate what cells type were involved in the fibrosis control. 

After activation, HSCs become proliferative and assume contractile and pro-inflammatory characteristics besides chemotactic capacity [117,119]. Activated HSCs produce chemokines such as MCP-1, CCL21, RANTES, and CCR5 and express TLRs, indicating their capacity to interact with DAMPs and PAMPs and, maybe, to amplify local inflammatory responses. HSCs also promote hepatic epithelial cells’ regeneration through the secretion of cytokines and growth factors as TGF-β and epithelial growth factor (EGF) and can revert to a quiescent state after the liver injury is resolved [127,128].

##### Liver Sinusoidal Endothelial Cells

The capillaries in the liver are formed by liver sinusoidal endothelial cells (LSECs), which comprise about 50% of the non-parenchymal cells. The fenestrae observed in LSECs vary from 100 to 150nm in diameter, freely allowing the diffusion of blood and stromal soluble factors. These cells can also actively transport drugs, polypeptides, and cationic components, for example, increasing the exchange of soluble factors in the organ [129]. Besides, LSECs are highly phagocytic and express multiple pattern-recognition receptors (PRRs) and scavenger receptors, such as stabilin-1 and -2, mannose receptors, and FcRs, that help clear antigens and potentially toxic molecules from the circulation. For example, in the liver, these cells are responsible for eliminating more than 75% of LPS that arrives from the intestines.

LSECs express low levels of MHC II and co-stimulatory molecules, like CD80, CD86, and CD40, and can cross-present antigens in MHC I molecules [130,131]. They also express CD54 (ICAM-I), CD105, and endothelial markers as CD106 (VCAM-I) and CD31. Most importantly, LSECs express Fas-L, PD-L1, and LSECtin (Figure 2), a molecule that recognizes CD44 on the surface of T lymphocytes and inhibits T cell activation, proliferation, and effector functions [132]. These cells also secrete low levels of IL-12 and, therefore, mostly lead to differentiation of Th2 over Th1 lymphocytes and Treg cells [117,133].

The LSECs are also involved in liver regeneration, and this capacity is at least partially due to the release of hepatocyte growth factor (HGF) and angiopoietin-2. Besides, the recruitment of bone marrow-derived progenitor cells occurs mostly through the secretion of vascular endothelial growth factor (VEGF) by LSECs [134]. In murine experimental infection, KC and LSECs are the main host cells observed with intracellular amastigotes.

### 1.6. The Liver as a Tolerogenic Organ

The liver is the only non-lymphoid organ that sustains T lymphocyte activation. However, these cellular interactions include a peculiar combination of regulatory molecules that subverts T lymphocyte priming and classical functioning. Furthermore, it is an organ with immunological characteristics of tolerance, with tissue architecture, molecular mechanisms, and cell populations prone to limiting immunity [135,136]. Therefore, the conventional concept of immunological danger recognition cannot be unequivocally applied to the hepatic environment.

The bias of hepatic immunity towards immune tolerance has long been known, but it was experimentally reinforced in the 1960s when allograft transplants in pigs were not rejected [137]. It was later observed that a previous liver transplant increased the chances of success when a second organ was grafted from the same donor [138,139]. The mechanism of this phenomenon is not understood, but we can speculate that the hepatic environment can instruct peripheral immunity to tolerate some molecules. 

Under steady-state conditions, most hepatic cell populations express and secrete low levels of co-stimulatory molecules and MHC plus peptide complexes, such as IL-12, IFN-γ, and TNF, for example. On the other hand, they express higher levels of IL-10, TGF-β, PGE2, and multiple inhibitory receptors, such as PD-1/PD-L1, Fas/Fas-L, cytotoxic T lymphocyte-associated protein 4 (CTLA-4), lymphocyte activating gene 3 (LAG-3), and T cell immunoglobulin and mucin domain-containing protein 3 (TIM-3) [139,140]. Moreover, the liver is considered a “graveyard” for T cells, a primary site for activated T lymphocyte apoptosis after cellular exhaustion [141]. Furthermore, CD8^+^ T lymphocytes can be deleted [142], CD4^+^ T lymphocytes usually differentiate into Th2 or Treg cells, and anergy, hyporesponsiveness, or activation-induced cell death (AICD) can be the outcome after T cell priming. Under such conditions, it is hard to imagine that a pro-inflammatory response could be orchestrated in the liver after infection by *T. cruzi* or by any other pathogen. 

### 1.7. Changing the Paradigm, the Trigger of Hepatic Immunogenic Responses

We have recently published evidence showing that the conventional recognition of danger molecules in secondary lymphoid sites and activated peripheral T lymphocytes are pivotal in altering intrahepatic cells’ phenotype and function after *T. cruzi* infection [46]. According to this view, activated peripheral T lymphocytes would instruct hepatic lymphoid and myeloid cells in the liver, changing the local balance of hepatic tolerance towards a pro-inflammatory response. Thus, under this new paradigm, peripheral T lymphocytes would exert supremacy over the hepatic environment.

Previous results obtained by other authors indirectly sustain this proposal. To date, after antibody-induced CD62-L neutralization, T lymphocytes were prevented from entering lymph nodes, and liver damage was no longer observed in transgenic mice with autoimmune hepatitis (AIH) [143]. However, the authors proposed that CD8^+^ T lymphocytes competed for the primary activation anatomical site, either in secondary lymphoid tissues or the liver. Then, according to this view, the initial anatomical activation site would determine the T cells’ fate as inflammatory or tolerant, and if the antigens were recognized in the liver, immune tolerance would be their defined response. It was also proposed that antigen-primed T lymphocytes migrate from the lymph nodes and are retained in the liver for activation when in the presence of a more pro-inflammatory environment, in the case of second antigen exposure [144]. Other authors showed that an IL-12-based vaccine reversed immune tolerance in the liver [145], and other papers showed the importance of the anatomical site of antigen expression in hepatic tolerance [146,147]. However, in all experimental approaches used, the possibility of antigen presentation in both lymph nodes and the liver could not be ruled out. Therefore, it is not possible to determine if activated peripheral T lymphocytes induced a pro-inflammatory response in the liver or if T lymphocytes primed in the liver itself acted locally [143,144,145,146,147].

It has also been suggested that high quantities of bacterial or viral molecules would be “an appropriate stimulation” for hepatic immunity [148,149]. However, commensal microbiota products in the liver are already very high under normal conditions [48], making this alternative implausible. On the other hand, it is interesting that systemic PAMPs can induce immunogenic responses in the liver, as observed in LPS-induced liver injury and LPS-deficient bacterial infection models [150,151]. Therefore, while incoming LPS that arrives from the portal vein are tolerated, circulating LPS can induce a robust hepatic response. We can speculate that when these molecules percolate lymph nodes and lead to peripheral T lymphocyte activation, the hepatic tolerance paradigm could be overthrown. 

### 1.8. The Hepatic Inflammatory Response in Trypanosoma cruzi Infection

Our group studied the *T. cruzi* infection and parasite antigen presentation in the context of hepatic tolerance and immunity and showed that peripheral activated T lymphocytes subvert the hepatic tolerogenic status [46,152].

When we orally administered *T. cruzi* extract to uninfected mice, whose PAMPs were never encountered by the hepatic tissue, the tolerogenic milieu was reinforced, with a significant increase of B7-H1, CTLA-4, IL-10, and TGF-𝛽 in the liver when compared with uninfected/untreated mice [46]. Conversely, intraperitoneal (i.p.) injection of *T. cruzi* extract induced a shift in the hepatic environment, with effector and effector memory T cells (Tem) expansion, PD1 down-regulation (unpublished data), and an increase of the pro-inflammatory cytokines TNF and IL-6 in the liver. 

We then hypothesized that the liver would mount a pro-inflammatory response once locally instructed by activated peripheral T lymphocytes, which passed the processes of central and peripheral tolerance and conventional danger recognition. To test this hypothesis and to avoid the interference of T lymphocytes activated in different compartments, including the liver, we performed adoptive transfer assays. We treated and boosted syngeneic donor mice with *T. cruzi* extract and sorted CD3^+^CD44^high^CD62^−^L^−^CD197^−^-activated splenic T lymphocytes. Then, recipient mice received a parasite extract by gavage and sorted T lymphocytes intraperitoneally. Compared with the control groups, and in the absence of in vivo infection, we observed a robust pro-inflammatory hepatic response after adoptive transfer of activated peripheral T lymphocytes. There were increased levels of TNF, IFN-γ, IL-6, and CCL2 in the liver and increased numbers of effector/Tem T lymphocytes and KC. These findings were accompanied by a reduction in Treg, NKT, and γδ T lymphocytes with increased liver damage. These results are summarized in Figure 3.

Using activated splenic T lymphocytes from GFP donor mice, we observed that most transferred cells were retained in the liver of recipient mice, and in vitro and in vivo assays are in course to discern what cell populations are involved in this cellular integration. 

We then evaluated if activated intrahepatic lymphocytes (IHL) play a systemic role in vivo after *T. cruzi* infection. GFP mice were i.p. treated with a parasite extract, and activated splenic T lymphocytes were purified and transferred to syngeneic wild-type recipient mice plus extract by gavage. Fifteen days after the transfer, recipients’ activated GFP^−^ IHLs were purified and transferred to new recipient mice immediately before infection [46]. The infected mice that received activated IHL had reduced blood parasitemia and increased skeletal muscle damage compared with infected mice that did not receive exogenous activated cells. These results indicate an inflammatory role of activated IHLs in systemic infection and that peripheral and hepatic immunity are two sides of the same coin that act in concert.

Much remains to be understood about the mutual signalings that mediate hepatic and peripheral cross-talks. These players compose the complex regulatory pathways of immunity and tolerance in organisms, with an enormous potential impact on therapeutic approaches for systemic and liver pathologies.

### 1.9. The Immunogenic Response in the Liver

Once the tolerance mechanisms are broken in the liver, pro-inflammatory pathways are established in the organ, activating multiple pathways that lead to immunity. Although most of the pathways described below were not yet studied in the liver after *T. cruzi* infection, they may also play a role in infection control.

As in the peripheral immune response, the liver can recognize two major classes of immunogenic molecules, which are DAMPs and PAMPs. In the first case, self molecules such as the high mobility group box 1 protein (HMGB1), fragments of extracellular matrix components or uric acid crystals are recognized as danger signals by receptors such as TLRs [153] and Nod-like receptors (NLR) [154]. PAMPs are pathogen-associated molecules, such as flagelin and double-strand RNA, and are recognized by receptors such as TLRs, NLRs, C-type lectin-like receptor (CLEC), and RIG-I-like receptors (RLR) [155].

When an immunological danger signal is sensed in the liver, and a pro-inflammatory response is triggered, LSECs become activated and signal leukocytes for trans-endothelial migration by mechanisms different from those observed in other tissues. In the liver, this is a selectin-independent process, a group of molecules required for leukocytes’ adhesion to peripheral postcapillary venules [156]. In vessels of peripheral non-lymphoid tissues, the leukocytes tether and rollover selectins and integrins expressed by both the activated endothelium and leukocytes while signaling in both directions.

In the sinusoids, cell adhesion is mainly based on a particular set of molecules that include vascular adhesion protein-1 (VAP-1), CD44, and hyaluronan [157,158,159,160]. However, activated LSECs may also mediate leukocyte transmigration after the engagement of selectins and integrins, such as VLA4 with VCAM-1 [158]. Cellular migration to the liver can also occur non-vascularly, and in this case, it is mediated by CD44 and ATP released from damaged cells [161].

### 1.10. The Role of Liver Cells in the Local Immunogenic Response

In the case of infections, LSECs can contribute to immunogenicity [141]. Murine cytomegalovirus infection can stimulate LSECs to present antigens to T lymphocytes, promoting the differentiation of effector CD8^+^ T lymphocytes independently of DCs [162]. This rapid activation of CD8^+^ T lymphocytes, with granzyme B expression, depends on IL-6 trans-signaling to make CD8^+^ T lymphocytes susceptible to IL-2 released from Th1 cells [163,164]. Moreover, LSEC-stimulated T lymphocytes can migrate to secondary lymphoid tissues in a CCR7-dependent manner and differentiate into effector T lymphocytes after interaction with TCR and co-stimulus via CD28 [165]. 

The liver also plays an important role in the generation of memory T lymphocytes capable of contributing to effector immune responses after stimulation by immunogenic HDCs (IL-12-producing cells) [165,166,167]. In mice, Tem lymphocytes are CD62L^−/low^CD44^high^CD127^+^CD197^−/low^; central memory T lymphocytes are CD62L^high^CD44^high^CD127^+^CD197^high^; effector T lymphocytes are CD62L^−/low^CD44^high^CD127^−^CD197^−/low^, and naïve T lymphocytes are CD62L^+^CD44^low/int^CD197^−^ [168,169]. Resident memory T lymphocytes were more recently described, whose phenotype is CD62L^low^CD197^−/low^CD44^high^CD103^+^ [170,171], although the expression of CD103 is questionable in the liver. Memory T lymphocytes can secrete cytokines, and Tem lymphocytes, for example, can produce IFN-γ [168,172]. After stimulation by APCs, such as HDCs, memory T lymphocytes can proliferate quickly and differentiate into effector T lymphocytes [165,173].

In the liver, myeloid cells stimulated by TLR and TNF, for example, can adhere to the hepatic sinusoids and form structures named intrahepatic myeloid aggregation for T cell expansion (iMATE). iMATE appears rapidly after infections along the sinusoidal vessels, stimulating the proliferation of CD8^+^ T lymphocytes and providing a hepatic microenvironment where T cells are not exposed to negative regulatory signals [174]. The iMATE-dependent T lymphocyte proliferation results in a 50–100-fold expansion of the CD8^+^ effector T lymphocyte population [175]. Therefore, iMATE formation is essential for immunogenic liver responses, as seen in viral infections [174] and hepatocellular carcinoma [176].

Regarding B lymphocytes, they can rapidly increase the expression of CD40, CD80, and CD86, and produce IL-12, IFN-γ, IL-6, and TNF after LPS stimulation in the liver. Besides, B lymphocytes can stimulate HDC to express CD86, increase the secretion of IL-6 and IL-12, and reduce IL-10, demonstrating that liver B lymphocytes also have pro-inflammatory properties in the organ [177,178,179].

#### 1.10.1. HDC, KC, HSC, and Cholangiocytes in the Hepatic Effector Response

Besides the main HDCs observed, an extraordinary and minor population of DC is also found in the liver, possibly an intermediate population with cytotoxic properties that can efficiently lead to T lymphocyte activation. These NK1.1^+^ cytotoxic HDCs, also found in other tissues, are inflammatory cells that can produce high levels of IFN-γ via autocrine IL-12 stimulation, leading to the development and activation of cytotoxic T lymphocytes [54,180,181,182]. 

Regarding KCs, they compose a heterogenic population in the liver [161,183,184,185]. These cells can express several receptors, including TLR, RIG-II, C-type lectin-like receptor (CLEC), and NOD, contributing to the induction of inflammation in the liver through inflammasomes. Subsequently, the release of IL-1β leads to an inflammatory response [186,187,188]. KCs can also cooperate with platelets via adhesion receptors, contributing to the control of bacterial infection [189]. Collaborative interactions between KCs and neutrophils are also important in anti-bacterial responses in the liver [190,191]. Moreover, KCs can present microbial antigens to CD8^+^ T lymphocytes or NKT cells, stimulating T lymphocytes proliferation and antimicrobial responses [192]. 

Activated HSCs express MHC I and II, CD1d, co-stimulatory molecules, and produce cytokines, as cited before. In addition, they can present antigens, inducing the activation of CD8^+^ T lymphocytes, CD4^+^ T lymphocytes, and NKT cells, respectively [121,122,193]. Moreover, HSCs can cooperate with LSECs by transferring MHC-I molecules and contributing to an effective antiviral response by stimulating CD8^+^ T lymphocytes activation [194].

Cholangiocytes can induce immunogenic responses using MHC class I-like-related molecule (MR1) to induce T lymphocyte activation in the human liver. Vitamin B metabolites released from both pathogenic and commensal bacteria can be recognized by mucosal-associated invariant T cells (MAIT) through MR1. Then, pro-inflammatory MAIT can secrete IFN-γ, TNF, IL-17, and granzyme B and protect the biliary tract from infiltrating bacterial infection [195,196]. In human cholangitis, IL-18 expression has been shown to be stimulated and led to the synthesis of pro-inflammatory cytokines that affected the epithelial integrity of cholangiocytes. This was also observed in an experimental model with NLR family pyrin domain containing 3 (NLRP3) overexpression in activated cholangiocytes [197,198,199].

#### 1.10.2. Lymphoid Cells in the Liver Immunity

As previously cited, MAIT cells observed in the context of liver infection can be activated in an MR1-dependent manner, present bacterial ligands, and express IL-12, IL-15, and IL-18 [200,201,202,203]. Moreover, after proper stimuli, MAIT cells express activation markers such as CD69 and CD38, produce IFN-γ, IL-17, and TNF, lead to pro-inflammatory stimulation of both KCs and cholangiocytes, and can release granzyme B. Thus, they exert cytotoxic activity against infected cells [200,201,204,205,206].

NK and NKT cells play important roles not only in antiviral, antimicrobial, and antitumor responses in liver lesions but also in liver fibrosis and repair. Both cell types can induce liver damage by IFN-γ production and hepatocyte’s death [95,207]. NK cells migrate quickly to the inflammation site, can be stimulated by cytokines such as IL-12, IL-15, and IL-18, as well as type I IFN [87], release cytotoxic granules containing perforin and granzymes, and promote virus-infected hepatocytes lysis [208]. NKT cells can be recruited to the liver by CXCR6 and, after stimulation, can secrete large amounts of IFN-γ, IL-4, GM-CSF, and other chemokines and cytokines [206]. NKT cells are involved in liver damage of various etiologies, including autoimmune liver injury, alcoholic liver disease [209], non-alcoholic fatty liver disease [210,211], as well as an LPS-induced injury [212]. Moreover, NKT cells can, for example, be activated by hepatitis B virus (HBV) and HCV via CD1d [94], inhibiting viral replication in hepatocytes by IFN-α secretion [213,214]. In this sense, they are involved in the effectiveness of IFN-α treatment in chronic HCV infection [215]. NKT cells can also be stimulated in cooperation with KCs, leading to NKT cell activation, with the production of IFN-γ and the control of bacterial infection, such as *Borrelia burgdorferi* [216].

In addition to NK and NKT cells, different subpopulations of γδ T lymphocytes have been described in the liver. Among them, we can highlight the CD95L^+^ and Vδ1^+^ subpopulations (both TNF and IFN-γ producers), besides CCR6^+^CD95L^+^, and Vγ4^+^ subpopulations (both produce IL-17) [217,218]. γδ T Vδ1^+^ liver lymphocytes have been described as necessary for the effective antitumor response in a TNF- and IFN-γ−dependent manner [219]. Such γδ T lymphocyte populations have been associated with immunogenic responses in HCV infection [220,221,222]. In plasmodium infection, γδ T lymphocytes were also described as having an important protective role [223,224].

#### 1.10.3. Neutrophils and Eosinophils in Liver Inflammation

After activation, neutrophils can follow a gradient of chemokines and migrate to the hepatic parenchyma. This chemotaxis involves the expression of integrins on neutrophils (as CD11a) and endothelial cells (as ICAM-1) [225]. Additionally, neutrophils can sequester bacteria, viruses, bacterial products, and even platelets by launching chromatin in extracellular traps named NET [226,227,228,229]. Regarding platelets, they can adhere to neutrophil’s surface and form large aggregates that contribute to virus elimination due to the recruitment of cytotoxic T lymphocytes to the liver [230]. Thus, platelet aggregation is increasingly recognized as an innate defense mechanism in the liver [231].

The involvement of eosinophils in liver inflammation and damage in animals and humans was demonstrated in hepatitis induced by concanavalin A, which relies mainly on NKT cells and IL-5 production [232,233,234,235]. It has also been shown that eosinophils can be recruited after hepatic necrosis and release IL-1β and IL-18 both in *Schistosoma mansoni* infection and sterile inflammation [236]. In halothane-induced liver injury, eosinophils were associated with leukocyte recruitment and tissue damage with increased serum transaminases activity [237]. The cytotoxic activity of eosinophils occurs mainly by releasing preformed cationic proteins, including major basic protein and eosinophil cationic protein [238].

Reversing or breaking hepatic immune tolerance in persistent infections or cancer is of central importance. This can be demonstrated by the reversion of liver T cell tolerance and viral persistence in the case of HBV infection after the blockade of inhibitory receptors, such as PD-1 [239,240]. Additionally, in tumors such as hepatocellular carcinoma, PD-1 blockade has good results in inducing a protective immune response [241,242,243]. 

## 2. Perspectives

Today we know many of the biochemical pathways and cell populations that sustain hepatic tolerance, and this knowledge should be translated into practical proposals and alternative treatments for autoimmune or chronic inflammatory diseases, for example. A few initiatives have been proposed, such as the induction of hepatic Treg cells by LSEC to control autoimmune encephalomyelitis (EAE) [244]. Previously, the authors showed that LSECs could induce CD4^+^ Foxp3^+^ Treg cells in mice [245]. Subsequently, they injected nanoparticles as carriers of autoimmune peptides in vivo, which were selectively delivered to LSECs. This treatment resulted in antigen-specific Treg cell induction, and these cells could completely and permanently prevent the onset of clinical EAE. Moreover, in mice with established clinical EAE, the treatment for Treg cell induction rapidly and substantially improved muscle paralysis and atonia, whereas the control group deteriorated. Similar results were obtained when antigen-specific Treg cells were expanded using neural autoantigen myelin basic protein (MBP) in the liver [246]. In the case of myelin infusion, antigen-specific Foxp3^+^ Treg cells exerted their effect by diminishing antigen-bearing inflammatory dendritic cells recruitment to lymph nodes and by impairing their function [247]. The induction of Treg cells in the liver or the periphery is reviewed in [248,249].

Other approaches such as those using antigen-specific immunotherapy [250] and neutralization of immunomodulatory molecules in the liver (reviewed in [251]) are being studied or are already in use. Much remains, however, to be understood about the fascinating pathways and immunological cross-talks hidden in the liver immunophysiology, including after *Trypanosoma cruzi* infection.

## Figures and Tables

**Figure 1 pathogens-10-01074-f001:**
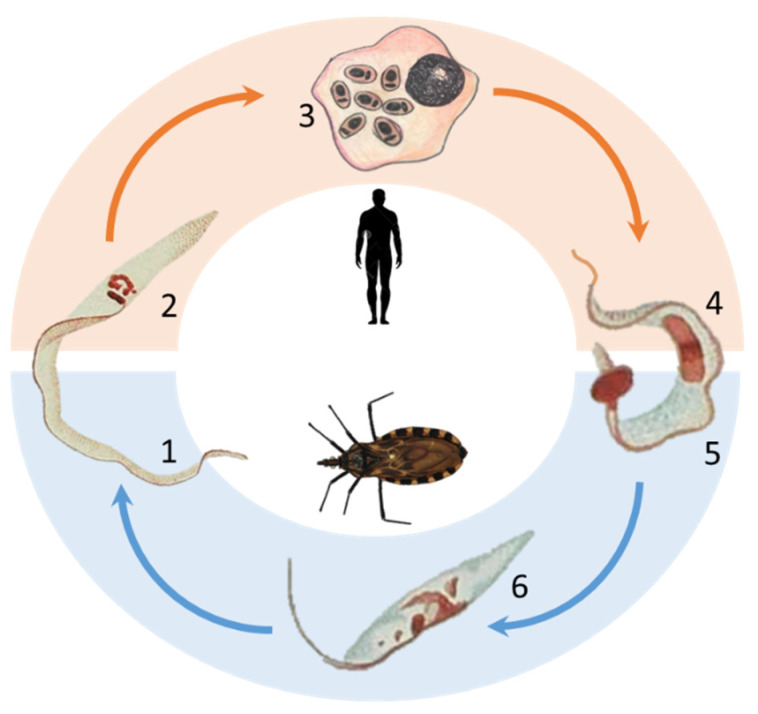
The *Trypanosoma cruzi* biological cycle. When the invertebrate host feeds on a human being, or any other vertebrate host, the trypomastigote metacyclic form (1) is transmitted. Then, this form (2) rapidly invades a host cell and differentiates in the cytoplasm into amastigote forms (3), which duplicate by binary divisions. Then, these forms intracellularly differentiate into trypomastigote forms that are released (4) to infect other host cells or be obtained (5) by the invertebrate host. The trypomastigote forms then start the differentiation into epimastigote forms (6) that adhere to the insect’s intestinal epithelium. These forms also proliferate by binary divisions until their differentiation into metacyclic trypomastigote forms, reinitiating the cycle. The epimastigote and trypomastigote forms were adapted from [3]. The lower segment in blue represents the invertebrate cycle, while the upper segment represents the vertebrate hosts’ cycle. The parasite forms are not presented to scale.

**Figure 2 pathogens-10-01074-f002:**
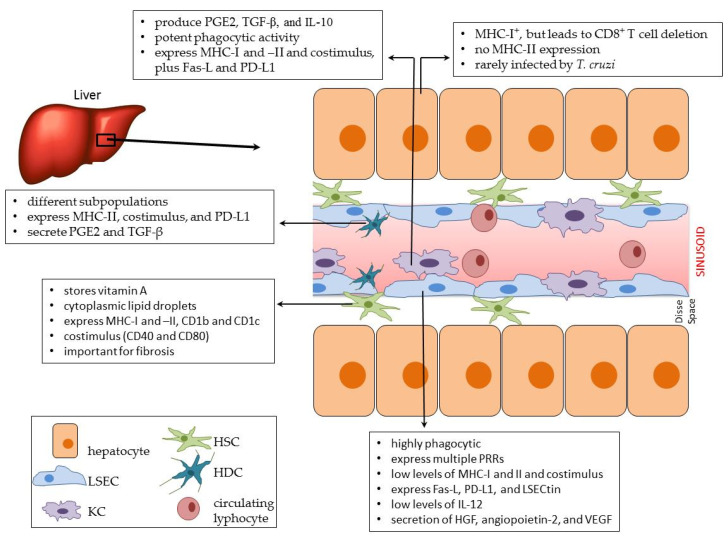
Liver microenvironment and resident cells under steady-state conditions. The portal vein delivers blood rich in molecules from the intestinal flora to the liver. This blood flows through the hepatic liver sinusoids lined by fenestrated liver sinusoidal endothelial cells (LSEC). The normal liver contains LSECs, hepatocytes, hepatic stellate cells (HSCs), hepatic dendritic cells (HDCs), lymphocytes, and Kupffer cells (KCs). Between the sinusoid walls and hepatocyte cords, there is the Disse space. The general characteristics of each cellular population are indicated in the boxes. HGF, hepatocyte growth factor; VEGF, vascular endothelial growth factor; PD-L1, programmed death-ligand 1; PRR, pattern recognition receptors; PGE2, prostaglandin E2.

**Figure 3 pathogens-10-01074-f003:**
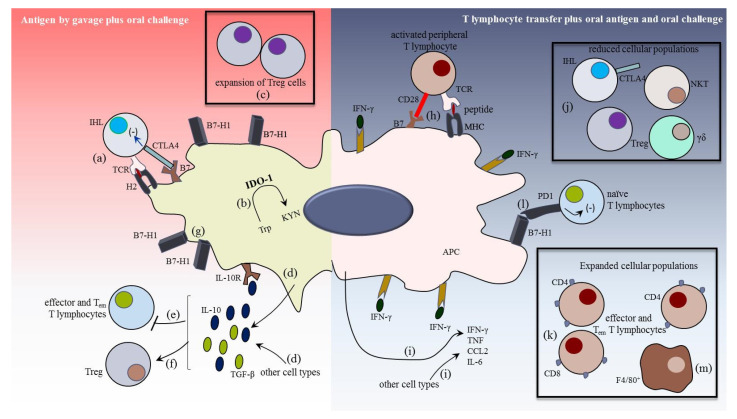
Proposed pathways for hepatic tolerance to portal antigens and the reversion to a “pro-inflammatory” milieu after activated peripheral T lymphocyte transfer. The left panel represents the group of mice that received a parasite extract by gavage, showing CTLA-4^+^ intrahepatic T lymphocytes interacting with hepatic B7-bearing antigen presenting cells plus antigen (**a**). There is also IDO-1 upregulation (**b**), and Treg cells increase in the liver (**c**). We observed high levels of IL-10 and TGF-β (**d**), reduced numbers of effector and/or effector memory (T_em_) T cells (**e**), and more Treg cells (**f**). Moreover, there was increased expression of B7-H1 in the liver stroma (**g**), a known down regulator of T lymphocytes’ function. The right panel represents the group that received parasite extract by gavage plus activated peripheral T lymphocytes by adoptive transfer. In this scenario, CTLA-4^−^ activated peripheral T lymphocytes would interact with APCs in the liver through the classical engagement of CD28/B7 and TCR/MHC plus peptide (**h**), and in this group, we observed an increased production of the pro-inflammatory cytokines IFN-γ, TNF, CCL2, and IL-6 (**i**). In this ambience, cells such as CTLA-4^+^ IHL, NKT, Treg, and γδ T lymphocytes (**j**) were reduced in the hepatic stroma, and CD4^+^ and CD8^+^ effector/T_em_ lymphocytes (**k**) were increased. In the presence of activated peripheral T cells, there is a balance between pro- and “anti-inflammatory” pathways, with increased expression of PD-1 and B7-H1 in the liver (**l**). Finally, there is an increase in F4/80^+^ cells in the liver (**m**), with still unknown functions. Adapted from [46].

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
