# Peer review of "The Liver and the Hepatic Immune Response in Trypanosoma cruzi Infection, a Historical and Updated View"

_pathogens, 2021, doi:10.3390/pathogens10091074_

Round 1

Reviewer 1 Report

The liver and the hepatic immune response in Trypanosoma cruzi infection.

The authors provide an accurate and enjoyable historical account of Chagas disease and its implications in liver metabolism, besides the updated and detailed data on the biochemistry of hepatic cells. To summarize the complex information provided in the text, I would suggest that the authors include a picture, drawing, or scheme of the liver with the cells described in the text. It would also be helpful to have a diagram of the cells with their main signaling molecules, perhaps at the side of the liver diagram. That should be before Fig. 2. Finally, a final thought to link the infection by T. cruzi with its impact on the liver could be added in the last paragraph. Last but not least, there are minor issues with the use of English that make the reading a bit hard. Some comments to improve the use of English are:

Line

Suggestion

38

Change “great numbers” for “many”

42

Change [2 with access link] for the reference

45

[3 with access link] should be a reference

65

Change “and” for “but”

66

parasite’s

66-7

…and all “of them” were described

86

“acknowledged” instead of “knowledged”

123

“in” instead of “on”

123

…mainly in countries “like” or “such as”

125

Change “are concentrated” by “remain”

132

Add a colon after “countries”

138

Add “such” before as

154

Add “vertebrate” before “host”

165

Add “mashed fruit” before “pulps”. “…mashed fruit pulps are part of the diet in the local culture”

165

Change “are” by “can be”

166

Change “more significant” by “higher”

279

Delete “of T. cruzi”

281

Delete “T. cruzi”

297

Delete “the” before T. cruzi

307

Change “others” by “other alterations”

315-16

Rephrase the sentence: “molecules of the intestinal microbiota”

330

Change “harbors” by “harbor” (the sinusoids are plural)

355

“C protein” instead of “protein C”

379

Rephrase by “When these cells mature, they down-regulate…”

387

Change by “apoptosis of neutrophils”

536

Add “to be” before “the most”

537

Remove “experimental” (a murine model is always experimental)

556

Rephrase by “… multiple patterns of recognition”

559

Change “the” by “that” before “arrives”

577

Rephrase by “classical function” and remove “notion”

585

Rephrase by “The mechanism of this phenomenon is”

740

Add a colon after “nodes”

741

Rephrase by “activation during…” and “in case of a second…”

745

Remove “the” before “liver”

773

Remove “in” before “liver”

775

Add “a” before “parasite extract”

780

“findings” instead of “finds”

783

Rephrase by “the liver of recipient mice”

786

Add “a” before “parasite extract”

789

Remove “in vivo” (the referred experiment in mice is in vivo)

800

“pro-inflammatory” (add “-“)

802

Add “a” before “parasite extract”

814

“anti-inflammatory” (add “-“)

884

Change “In regards” by “Regarding KC”.

887

Rephrase by “Subsequently, the release of IL-1b leads to an inflammatory response”

913

Rephrase by “Thus, they exert…”

917

Remove colon after “damage”, and add the apostrophe after “hepatocytes”, or change for “death of hepatocytes”. Perhaps the authors can specify whether the death of hepatocytes occurs by apoptosis. If that is so, change “death” by “apoptosis”.

Author Response

Dear Editor

We believe that the manuscript was substantially improved after the reviewers’ comments, critics, and suggestions. We do hope that this prestigious Journal can accept the revised version. Please, feel free to contact us if the issues raised by the reviewers were not adequately addressed in this version; we will be more than happy to make further adjustments.

Rveiewer #1

To summarize the complex information provided in the text, I would suggest that the authors include a picture, drawing, or scheme of the liver with the cells described in the text. It would also be helpful to have a diagram of the cells with their main signaling molecules, perhaps at the side of the liver diagram. That should be before Fig. 2.”

Dear editor, we prepared a new Figure 2 summarizing the anatomical distribution of the main hepatic cells and their highlights under steady-state conditions.

Finally, a final thought to link the infection by T. cruzi with its impact on the liver could be added in the last paragraph.”

We understood that the last paragraph was not mentioning the T. cruzi infection, the focus of our manuscript. However, there are no virtually experimental data in this regard, and any suggestions or commentaries would be highly speculative. Therefore, to meet this demand and pertinent comment, we added in the last phase the following statement: “Much remains, however, to be understood about the fascinating pathways and immunological cross-talks hidden in the liver immunophysiology, including after Trypanosoma cruzi infection.”

Last but not least, there are minor issues with the use of English that make the reading a bit hard. Some comments to improve the use of English are:”

All alterations were included, and we made a complete revision of the text to correct grammatical and syntax mistakes. However, some suggestions were altering the primary idea and content; in these cases, we modified the sentences and rephrased them to make the statements more straightforward.

Dear editor and reviewers, and are deeply thankful for all suggestions and comments. We believe that the manuscript is improved and do hope that we satisfactorily addressed all required corrections.

Very kind regards

Dr Andrea Henriques-Pons

Reviewer 2 Report

Dear Authors:

 The authors have carried out a review tittled “The liver and the hepatic immune response in Trypanosoma cruzi infection. A historical and updated view”.

In this review the authors describe the liver’s participation in the infection and the ones that followed in the last century until recently published results.

 This manuscript is a very enriching reading and well raised from a conceptual and academic point of view associated with a scientific rigor according to the standards of the journal.

Some considerations need to be taken into account:

  • The introduction is excessively long
  • As a suggestion to facilitate the reading of the manuscript, I would recommend to separatethe introduction in a historical introduction of the disease and another different section with the update of the main topics as it is defined in the title of the manuscript
  • Excessively dense manuscript that would require some more illustrations, figures or tables to make it more friendly
  • Lymphoid cells in the liver immunity (Line 908) ¿should it be classified as 1.9.2?
  • Neutrophils and eosinophils in liver inflammation (line 939) ¿should it be classified as 1.9.3?
  • Some of the references must be updated and its writing must be homogeneous (As an example 109,185,241)

Kind regards

Author Response

Dear Editor

We believe that the manuscript was substantially improved after the reviewers’ comments, critics, and suggestions. We do hope that this prestigious Journal can accept the revised version. Please, feel free to contact us if the issues raised by the reviewers were not adequately addressed in this version; we will be more than happy to make further adjustments.

Reviewer #2

 Some considerations need to be taken into account:

“The introduction is excessively long”

Although we agree with this comment, we would like to ask to keep the text how it was conceived and presented. We sought to contribute to interested readers with a complete picture of the field, starting with the very first mention of the liver involvement in the description of the Chagas disease and evolving to what we know today about each cell population under steady-state conditions and what we know about the impact of the infection over the organ and its cellular components. We believe that removing any parts of this work would make it incomplete and not entirely adequate to our purpose. Therefore, if this point is a suggestion and not a final recommendation, we would like to keep the paper’s design the way it is currently presented.

As a suggestion to facilitate the reading of the manuscript, I would recommend to separate the introduction in a historical introduction of the disease and another different section with the update of the main topics as it is defined in the title of the manuscript

As a review manuscript with no experimental figures, we found it difficult to separate the text into sections, as indicated by the reviewer. Therefore, we further indicated subheadings that were accordingly numerated. We do hope that this alteration will meet the reviewer’s indication.

Excessively dense manuscript that would require some more illustrations, figures or tables to make it more friendly.”

We agree and included a new Figure 2 summarizing the main hepatic cell populations and their highlights besides their anatomical location.

Lymphoid cells in the liver immunity (Line 908) ¿should it be classified as 1.9.2?

Neutrophils and eosinophils in liver inflammation (line 939) ¿should it be classified as 1.9.3?”

It was updated and corrected

Some of the references must be updated and its writing must be homogeneous (As an example 109,185,241)

Indeed there were many inconsistencies, and we corrected them all. Although we used the endnote software, many important information was missing, unformatted, or unnecessarily included (like DOIs).

Dear editor and reviewers, and are deeply thankful for all suggestions and comments. We believe that the manuscript is improved and do hope that we satisfactorily addressed all required corrections.

Very kind regards

Dr Andrea Henriques-Pons